# Using the GeoWEPP Model to Predict Water Erosion in Micro-Watersheds in the Brazilian Cerrado

Wellington de Azambuja Magalhães [1,*], Ricardo Santos Silva Amorim [2], Maria O'Healy Hunter [1], Edwaldo Dias Bocuti [1], Luis Augusto Di Loreto Di Raimo [1], Wininton Mendes da Silva [3], Aaron Kinyu Hoshide [4,5] and Daniel Carneiro de Abreu [5,6]

1   Curso de Doutorado Programa de Pós-Graduação em Agricultura Tropical, Universidade Federal de Mato Grosso, UFMT, Cuiabá 78060-900, MT, Brazil
2   Departamento de Engenharia Agrícola, Universidade Federal de Viçosa, DEA/CCA/UFV, Viçosa 36570-000, MG, Brazil
3   Empresa Mato-Grossense de Pesquisa, Assistência e Extensão Rural (EMPAER-MT), Centro Político Administrativo, Cuiabá 78049-903, MT, Brazil
4   College of Natural Sciences, Forestry and Agriculture, The University of Maine, Orono, ME 04469, USA
5   AgriSciences, Universidade Federal de Mato Grosso, Caixa Postal 729, Sinop 78550-970, MT, Brazil
6   Instituto de Ciências Agrárias e Ambientais (ICAA), Universidade Federal do Mato Grosso, Campus Universitário de Sinop, Avenida Alexandre Ferronato, 1200, Sinop 78550-728, MT, Brazil
*   Correspondence: wellingtonagro@gmail.com; Tel.: +55-65-99961-7964

**Abstract:** The GeoWEPP model has estimated water and soil losses caused by erosion at the watershed level in different parts of the world. However, this model was developed and its parameters have been adjusted for temperate climates, which are different from tropical climates such as those found in Brazil. Our study evaluated the performance of the GeoWEPP model in estimating soil erosion in three micro-watersheds in the Cerrado (i.e., savannah) of southeastern Mato Grosso state, Brazil. Major land uses modeled were soybean and corn cultivation, traditional pasture, and native vegetation. Input parameters for the GeoWEPP model involved climate, soil, land use and management, and topography. GeoWEPP was calibrated with input parameters for soil erodibility specified as interrill and rill soil erosion, soil critical shear stress, and saturated hydraulic conductivity obtained experimentally and estimated by internal routine equations of the GeoWEPP model. Soil losses observed in micro-watersheds with agriculture, pasture, and native vegetation were 0.11, 0.06, and 0.10 metric tons per hectare per year, respectively. GeoWEPP best modeled soil erosion for native vegetation and pasture, while over-estimating that for crops. Surface runoff was best modeled for crops versus native vegetation and pasture. The GeoWEPP model performed better when using soil erodibility input parameters.

**Keywords:** environmental impact; soil conservation; water erosion prediction; watershed; WEPP parameters

## 1. Introduction

Land use and occupation have undergone several transformations in recent decades, especially in emerging countries such as Brazil, as well as more globally. Studies on soil erosion have been essential for the management and conservation of the environment in general, especially in agricultural environments with higher erosion potential. Prior research has estimated soil loss in the order of 820 million metric tons per year in Brazil, considering not only annual crops but areas cultivated with pastures and perennial crops [1]. Another study estimated soil losses in Brazil to be around 616.5 million metric tons per year with annual crops alone [2]. Water erosion and nutrient runoff into water bodies are typical problems of agricultural activities [3]. Accelerated soil erosion has a global effect, and impacts on the emission of greenhouse gases such as carbon dioxide ($CO_2$), methane

($CH_4$), and nitrous oxide ($N_2O$) were responsible for 10 to 15% of global anthropogenic emissions in 2019 [4].

Erosion is the main cause of soil degradation in Brazil and the world, and studies aimed at minimizing the adverse impacts of erosion by guiding producers and technicians are increasingly important. Obtaining regional data, as well as methodological approaches, facilitates the adoption of sustainable agricultural practices, especially in biomes such as the Cerrado (i.e., savannah), where agricultural systems have been intensified in recent years in Brazil. Due to the great diversity and complexity of interactions of factors that govern the erosion process, which include combinations of climate, topographic relief, vegetative cover, and soil hydraulic properties, modeling is an essential tool to obtain both a quantitative and consistent approximation of soil erosion process and sediment transport rates under a wide variety of conditions [5]. In addition, such modeling can be used to define the most appropriate land use and management for each location [6–8].

One type of soil erosion model is GeoWEPP, which has been applied in several locations around the world to estimate water and soil loss [9]. This model was developed and calibrated for temperate climate conditions, which are very different from tropical climates. Therefore, developing research that aims to calibrate GeoWEPP to Brazilian edaphoclimatic conditions before being extensively used for erosion prediction is of paramount importance. In Brazil, some studies have already been developed using the WEPP model to estimate soil and water losses under certain soil management conditions. However, most of these studies have used this model to estimate soil losses without comparing model estimates with experimental data [10–13].

The Cerrado and southern Amazon regions in Brazil are characterized by intensive land use for agriculture and livestock. The micro-watersheds used in the present study predominantly had three types of land use. There were native vegetation, pasture, and annual cultivation of commodity crops (Figure 1). Major commodity crops in the region include soybean (*Glycine max* L.) during the wet season, followed by corn (*Zea mays* L.) in the wet/dry seasons (Santa Fe system). The growing of corn or corn under-seeded with pasture grasses such as *Brachiaria* sp. immediately following soybeans has become more prevalent in Brazil. In addition to these cropping sequences involving corn, soybeans can also be followed immediately by cotton (*Gossypium* sp.) as a sustainable intensification strategy to increase crop production on the same land area [14]. Although there has been a shift from conventional to no-till systems for soybeans, corn, and cotton, which can reduce erosion but increase reliance on agro-chemicals such as glyphosate (i.e., Roundup®) [15], conventional tillage is still used on ~40% of Brazil's total crop area [16]. Therefore, it is still important to validate soil erosion models for Brazil to complement field research in conventional tillage.

The GeoWEPP model is based on the physical principles of the erosion process, showing applicability in the simulation of the erosion process for different edaphoclimatic conditions [17]. It is essential that the model be calibrated for the Brazilian edaphoclimatic conditions, since the development of a model is quite costly in terms of time and necessary resources, due to both data collection and the application of guidelines involved in the process [18,19]. The study and application of tools such as GeoWEPP in developing countries such as Brazil can facilitate the mapping of areas sensitive to erosion. This can identify areas better suited for re-seeding perennial crops (e.g., pasture), Amazon re-forestation, and/or re-habilitation of Cerrado biome savannahs and other biomes in Brazil.

The development of this work is a sequence of studies and results that were being carried out in the micro-basins, involving hydrological monitoring techniques and mathematical modeling of sediment production and surface runoff. This study aimed to evaluate the performance of the GeoWEPP model in estimating sediment production when applied to three micro-watersheds of the Cerrado in Mato Grosso state, Brazil. These micro-watersheds are characterized by agricultural production of soybeans (*Glycine max* L.), maize (*Zea mays* L.), extensive pasture (*Brachiaria* sp.), and native vegetation. The specific

objectives of our research were to (1) evaluate the dynamics of sediment production related to different land uses over the period of study between 2013 and 2015 and to (2) determine the accuracy of the GeoWEPP model in predicting soil losses when applied to these three watersheds characterized by different soil types and land use.

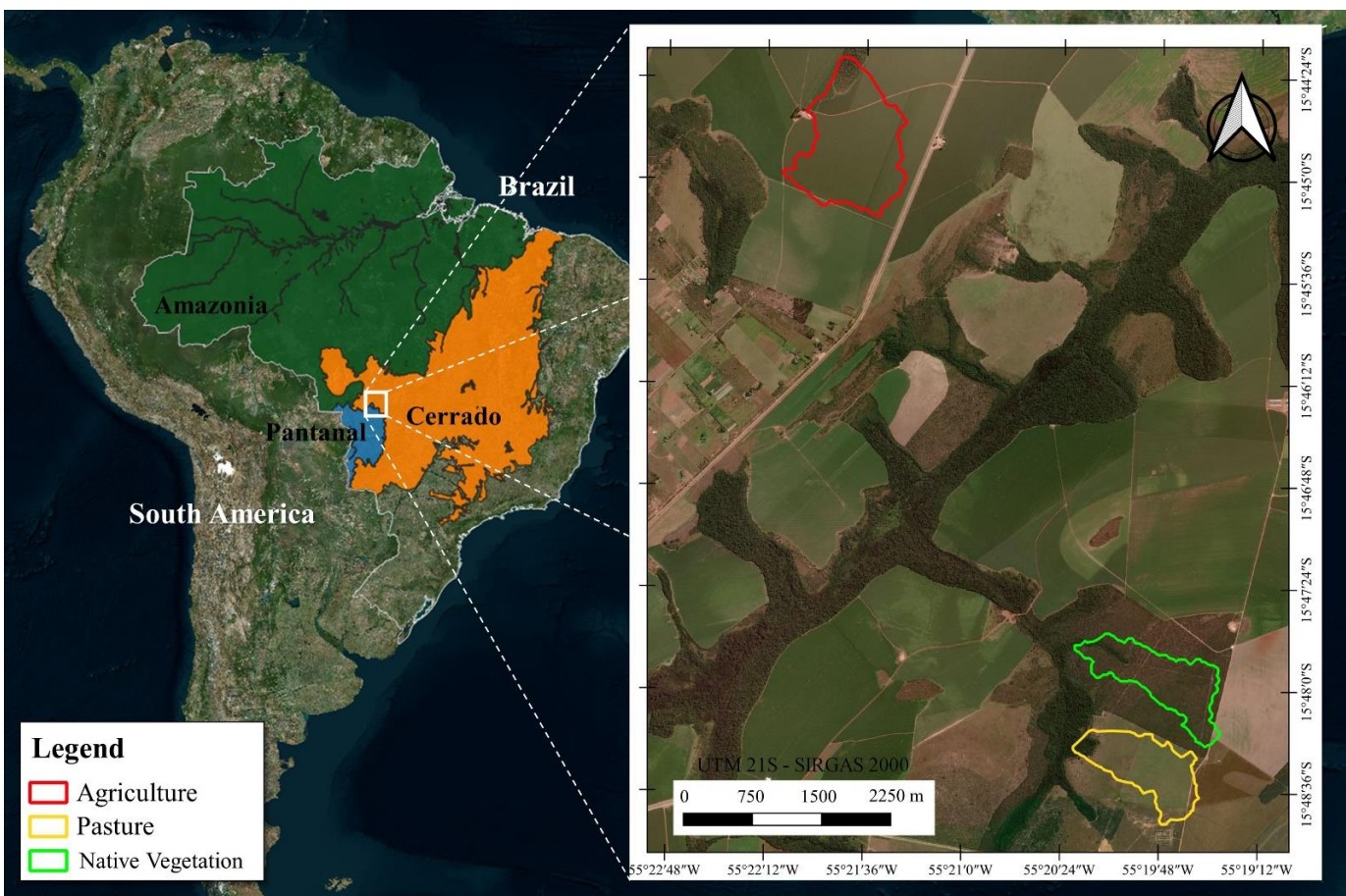

**Figure 1.** Location of study areas in Campo Verde, Mato Grosso state, Brazil.

## 2. Materials and Methods

### 2.1. Study Area

Our study was carried out in three micro-watersheds located in the municipality of Campo Verde, Mato Grosso state, Brazil (Figure 1). The research site is located in the Cerrado (i.e., savannah) biome in Brazil. These micro-watersheds are located within the Rio das Mortes watershed, one of the main tributaries of the Araguaia River. The regional climate is classified as tropical Aw, according to the Köppen classification, alternately wet and dry, with temperatures ranging between 18 and 24 °C. The mean annual precipitation is 1400 mm, varying between 800 and 1600 mm, with 70 to 80% of annual rainfall falling during the rainy season (October to May). The rainy season occurs from October to April and the dry season from May to September. The research area is located in the extreme northwest of the Paraná basin, belonging to the Cachoeirinha Formation. The lithology of the area has unconsolidated gravels, practically those that are monomictic, composed of >90% quartz pebbles that are well rounded with high sphericity. This formation is also characterized by sandy and clayey lenses, some strongly cemented by iron oxides. The reddish to yellowish colors from limonitization often form true laterized crusts [20]. These micro-watersheds were selected because they represented the most common forms of land use in Brazil's Cerrado biome, namely agricultural crops and pasture, as well as native vegetation kept as reserves (Figure 1).

The three selected micro-watersheds had specific characteristics for each of the three predominant land use types. First, commodity agriculture involved annual soybean and corn cultivation under direct sowing in the Santa Fe system for at least 10 years. Second, pasture was extensive with traditional livestock such as Nelore beef cattle (*Bos indicus*) with a stocking density of one animal unit (AU = 450 kg) per hectare. Finally, native vegetation was regenerated in native Cerrado forest, also more than 10 years old. This type of native vegetation is considered to be a reference for an environment that has not been altered for agricultural use.

## 2.2. Soil Characterization and Soil Sampling

The existing soil classes in the micro-watersheds are Latossolos (Oxisols) in agricultural areas and Neossolos (Entisols) for both pasture and native Cerrado, as defined by the Brazilian Soil Classification System [21]. Our study was carried out in two stages. The first stage involved sampling of soils in the three micro-watersheds. To carry out the experimental analyses of soil and water loss, three representative sampling points were selected within the watersheds. One point under pasture use, one under agricultural use, and one point with native vegetation. The criteria used for determining sampling locations were local relief, soil color and texture, history of use, and occupation. In each of the watersheds, soil samples were collected undisturbed using an auger and a kopeck ring (from the Soltest company). Soil was sampled down to three layers in the upper soil profile at 0 to 0.20, 0.2 to 0.4, and 0.4 to 0.6 m depth in a sampling grid of 200 × 120 m. The samples were sent for analysis to the Soil Physics Laboratory at the Faculty of Agronomy and Animal Science (FAAZ) at the Universidade Federal de Mato Grosso (UFMT) campus in Cuiabá, Mato Grosso state, Brazil. A more detailed description of soil characteristics in these watersheds can be found in Bocuti et al., 2016 [22] and Nóbrega et al., 2020 [23].

These data, associated with the physiographic characterization of the watersheds, were used to run the GeoWEPP model. The data collected also describe the morphometric characteristics and hydrological dynamics of these watersheds [18]. The second stage of our study involved physical and physical–hydric characterization of the study areas with the determination of the effective hydraulic conductivity ($K_e$), interrill erodibility ($K_i$), and rill erodibility ($K_r$) to calibrate the GeoWEPP model. The methodology proposed by the Agricultural Research Service of the United States Department of Agriculture (USDA) Agricultural Research Service (ARS) was used to determine $K_e$, $K_i$, and $K_r$ [24] for modeling by the GeoWEPP Software [9] compatible with the ArcGIS 10.4 version.

## 2.3. Soil Erosion Field Measurements

Between 2014 and 2016, our research team went to the experimental field site to determine the calibration parameters for rill erodibility (Kr) and soil critical shear stress ($\tau_c$) (Figure 2), in addition to the determination of effective hydraulic conductivity (Ke) and interrill erodibility (Ki) parameters (Figure 3). The furrow erodibility (Kr) and the critical shear stress ($\tau_c$) of the soil were determined from tests with the application of different flows in preformed furrows in freshly prepared soil with 9 m of length and 0.46 m of depth, with width and slope equal to the site conditions (Figure 3). During the tests, the surface velocity of the flow and the discharge rate of the furrows were measured.

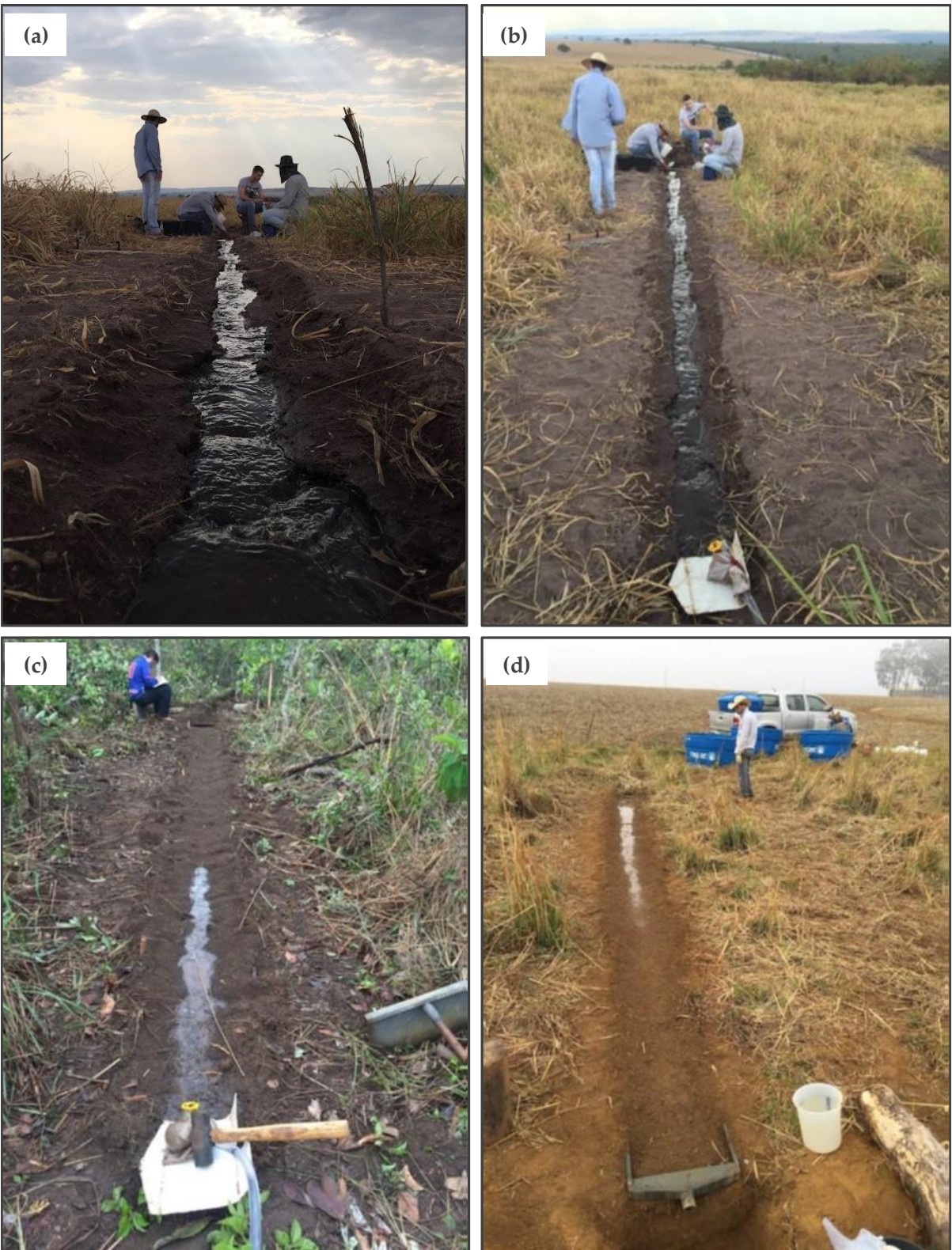

**Figure 2.** Preparation of areas for determination of erodibility in the furrow (Kr): (**a**,**b**) in the watershed under pasture, (**c**) in the micro-basin under native vegetation, and (**d**) in the micro-basin under agriculture.

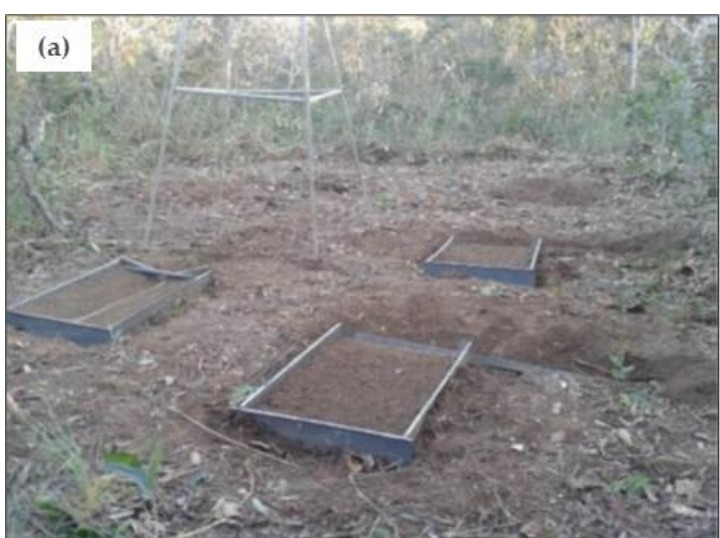
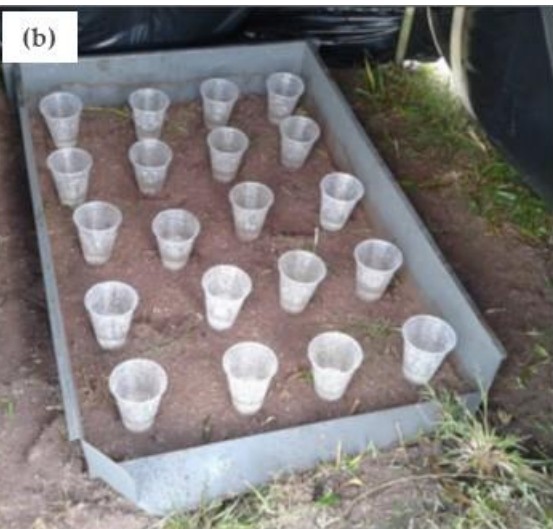

**Figure 3.** Installation of limiters in the Cerrado micro-basin (**a**) and determination of the average precipitation intensity (**b**) applied at the end of the Ke tests.

The effective hydraulic conductivity (Ke) of the soil was determined in the five study areas with three replications (two in the watersheds with agriculture, two in the pasture, and one in the native vegetation). We used a rainfall simulator following methods outlined in Bocuti et al., 2019 [25] (Figure 2). Interrill erodibility (Ki) in each area was determined in three experimental plots demarcated with galvanized sheets, following the methodology proposed by Elliot 1989 [23]. In the experimental plot, artificial rain was applied, and the Ki was calculated from the equation proposed by Foster 1982 [26] to estimate the sediment release rate in the interrill areas.

Surface runoff samples were also collected to determine the amount of suspended soil particles. Immediately after the start of surface runoff and when the flow rate became approximately constant, the geometry of the furrow and the height of the flow were determined. These data made it possible to determine the wetted area and perimeter, and consequently the hydraulic radius of the cross-section of the flow. For each type of soil in the studied areas, a graph of the average soil shear rate versus the average shear stress of the flow was generated, and a linear equation was adjusted to the set of points. The critical shear stress of the soil was considered to be the one in which soil loss was zero. Soil erodibility was obtained by using the slope of the trend line.

*2.4. GeoWEPP Setup and Application*

GeoWEPP integrates the WEPP model and the Topography ParameteriZation (TOPAZ) software into the ArcGIS 10.4® software [27] to predict sediment production and surface runoff at a watershed scale. The necessary input files for climate, slope, soil, and land use and management were generated in WEPP, and the topographic data were parameterized by TOPAZ based on a digital elevation model. A survey of altimetry was completed in the study region to delineate the catchment area of each basin using a precision GPS (TOPCON RTK Hiper Lite). With this survey, a digital elevation model (DEM) was generated, with $5 \times 5$ m pixels. Finally, the watershed was generated by GIS functions in ArcInfo, which is the command-line sub-package of ArcGIS. A database on WEPP extensions had to be structured for both modules. In general terms, the database was segmented by climate, soils, land use and management, and topography based on methods outlined by Maalim et al., 2013 [28]. A flowchart outlining all the geo-spatial processing steps is shown in Figure 4.

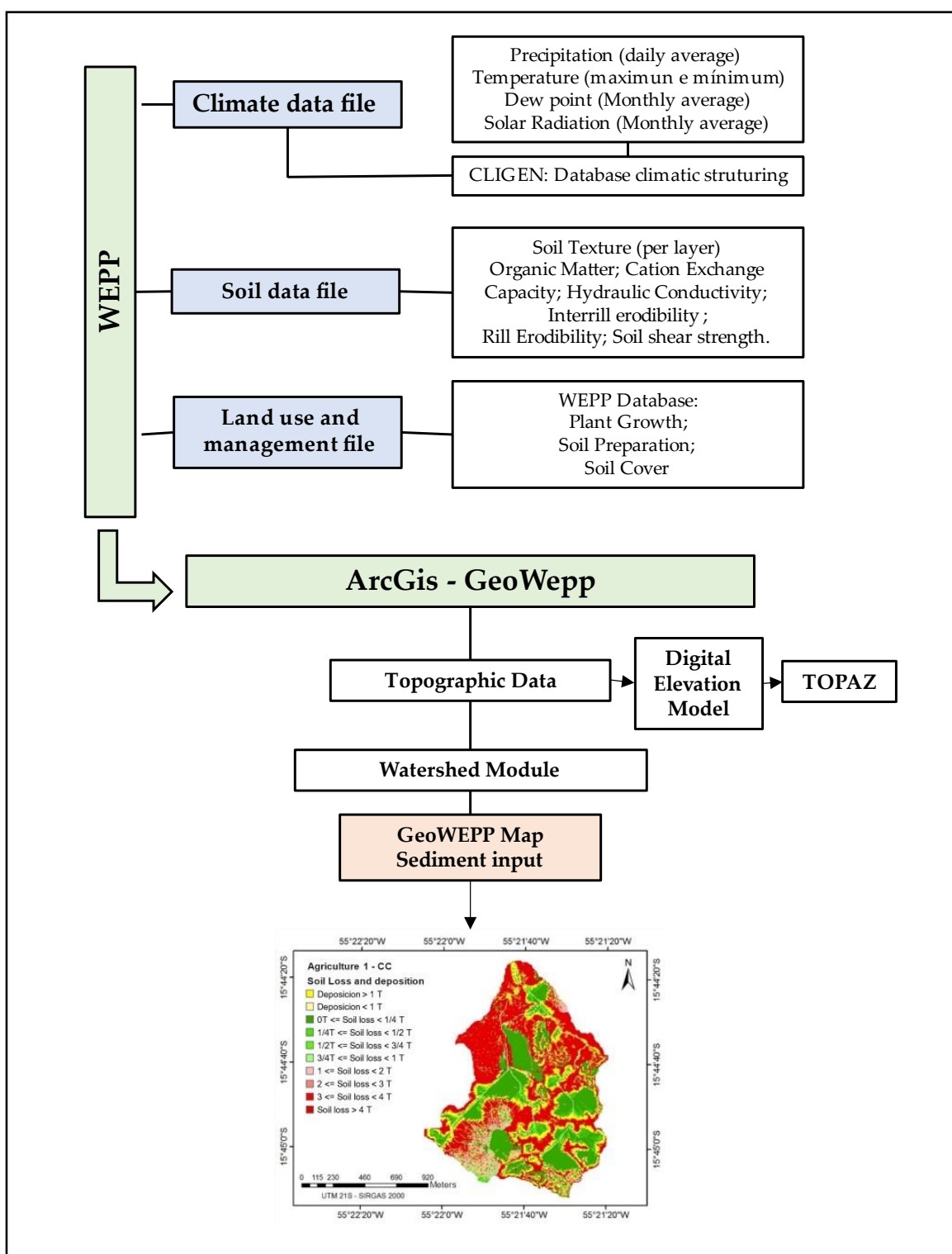

**Figure 4.** Methodology flowchart for modeling and preparing soil loss maps for different environments studied.

## 2.5. WEPP Input Data Preparation

### 2.5.1. Climate File

The daily input climate data file for the GeoWEPP model was generated using the CLIGEN software, version 4.3 [29]. The statistical parameters of precipitation, temperatures, and solar radiation were obtained from climatological stations we installed near the

study areas. These climatological stations were connected to data loggers that stored the information between 2012 and 2015 at 10 min intervals.

### 2.5.2. Topography and Land Cover

The digital elevation model (DEM) of the study watersheds was prepared using the "Topo to raster" tool in ArcGIS v.10.1, with a spatial resolution of $5 \times 5$ m. GeoWEPP integrates the WEPP model and the Topography ParameteriZation (TOPAZ) software into the ArcGIS $10.4^{®}$ software to predict sediment production and surface runoff at a watershed scale. The necessary input files for climate, slope, soil, and land use and management were generated in WEPP. The topographic data were parameterized by TOPAZ based on the DEM. Finally, the watershed was generated by GIS functions in ArcInfo. The digital elevation model (DEM) of the study watersheds was prepared using the "Topo to raster" tool in ArcGIS v.10.1, with a spatial resolution of $5 \times 5$ m.

GeoWEPP integrates the WEPP model and the Topography ParameteriZation (TOPAZ) software into the ArcGIS $10.4^{®}$ software to predict sediment production and surface runoff at a watershed scale. The necessary input files for climate, slope, soil, and land use and management were generated in WEPP. The topographic data were parameterized by TOPAZ based on the DEM. Finally, the watershed was generated by GIS functions in ArcInfo.

### 2.5.3. Soil File

The soil file that composes the physical attributes such as texture, soil saturated hydraulic conductivity, interrill erodibility, rill erodibility, and critical shear stress were generated by the internal routine equations of the WEPP model. Subsequently, the data determined in the field were inserted into the model, and new files were generated to be modeled in the GIS GeoWEPP interface. The experimental works were carried out in each of the pedological units of the micro-watersheds contemplated in the present study. Plots to determine hydraulic conductivity ($K_e$), interrill erodibility ($K_i$), rill erodibility ($K_r$), and critical shear stress ($\tau_c$) were installed for each pedological unit, with three replications performed for each parameter, following the WEPP methodology proposed by the United States Department of Agriculture (USDA) Agricultural Research Service (ARS) [23] and by [30]. Conventional soil tillage in the area consisted of removing existing plant residues followed by one plowing and two harrowing operations.

### 2.5.4. Management File

The land use and management file was generated from the management module integrated into the WEPP interface. The model program files were compiled based on local information available at each micro-watershed (native vegetation, pasture, and agriculture). Other input parameters were used from the WEPP database [24]. Thus, the output files were generated for each cropping system based on time frames that can be set by the user [9], which in our case was over the three years of field data collected.

### 2.5.5. Precipitation, Surface Runoff, Runoff Coefficient, and Soil Losses in Micro-Watersheds

Total precipitation and its temporal distribution were recorded in a pulse rain gauge installed in the experimental area and connected to a datalogger that stored the number of pulses in the collector every ten minutes. This enabled the evaluation of soil and water losses due to the different durations and intensities of precipitation. Triangular metal spillways were installed in each micro-watershed control section to allow all water in the course to pass through an area with known dimensions. The height (h) of the water depth that passes over the spillway sill needs to be known in order to calculate the flow rate.

A Hydrolab DS5X multiparameter probe, capable of estimating the hydraulic head of the course as a function of the hydrostatic pressure and determining the amount of sediment in suspension from the field calibration curve, was used to constantly monitor

the hydraulic head above the spillway sill. The probes were installed two meters upstream of the spillway so that the measured hydraulic head would not be influenced by the proximity of the spillway, where the water depth is lower due to the energy line. Height measurements were performed every ten minutes with data stored in the instrument itself.

An automatic water sample collector (Hach-Lange IL 2000-D with 24 bottles) was installed in each flow rate monitoring section whenever there was a rain event. The samples were used to determine the concentrations of sediments by the direct method and were used to prepare the calibration curve of the turbidity sensor. Field collections were carried out every 15 and 30 days during the rainy and dry seasons, respectively. The Kindsvater–Shen equation and its respective calibration adjustment functions were used to quantify the flow of each micro-watershed.

### 2.6. Statistical Analysis and Modeling

GeoWEPP performance and accuracy were evaluated by comparing the annual amounts of soil loss observed in the field with those estimated by the model and specific rainfall events. Data analysis and modeling were carried out using GeoWEPP both with and without calibration. When calibration was used, the maps of soil loss and surface runoff of the micro-watersheds were generated by inserting the Ki, Kr, Ke, and τc data determined within the micro-watersheds. During calibration, all other components such as climate, soil, land use and management, and topography were also inserted into the model. If calibration was not used, maps of soil loss and surface runoff of the micro-watersheds were generated without the insertion of Ki, Kr, Ke, and τc data. Under this condition, the model itself estimated the parameter values to generate the maps. Thus, only the digital elevation map and the climate, soil, land use and management, and topographic data were inserted into the model.

The evaluation of the efficiency of models was performed using the following three statistics. The first statistic was the root mean square error (RMSE):

$$RMSE = \sqrt{\frac{\sum_{i=1}^{n} \left(Y_{obsj} - Y_{estj}\right)^2}{n}} \tag{1}$$

where $Y_{obs}$ is the observed value, $Y_{est}$ is the estimated value, and $n$ is the total number of pairs of observed and estimated values for the ith treatment and *j*th repetition. The second statistic was the Willmott concordance index (*d*):

$$d = 1 - \frac{\sum_{i=1}^{n} \left(Y_{estj} - Y_{obsj}\right)^2}{\sum_{i=1}^{n} \left(\left|Y_{estj} - Y\right| + \left|Y_{obsj} - Y\right|\right)^2} \tag{2}$$

where *d* is the concordance index, and *Y* is the mean of observed values. The third statistic used was the Nash–Sutcliffe (*NS*) coefficient:

$$NS = 1 - \frac{\sum_{i=1}^{n} \left(Y_{obsj} - Y_{estj}\right)^2}{\sum_{i=1}^{n} \left(Y_{obsj} - Y\right)^2} \tag{3}$$

with similar parameters as RMSE and the Willmott concordance index.

## 3. Results and Discussion

### 3.1. Soil Characterization

Table 1 shows the soil physical attributes of the studied micro-watersheds. All three micro-watersheds with pasture and native vegetation have the same type of soil in common, Quartzarenic Neosol. This soil type is characterized by sand in the soil profile. The micro-

watershed with agriculture had a higher clay content with an oxidic characteristic, which is common in tropical soils.

**Table 1.** Soil attributes at a depth of 0 to 20 cm within agriculture, pasture, and native vegetation micro-watersheds in the municipality of Campo Verde, Mato Grosso state, Brazil.

| Soil Property | Unit | Agri-Culture 1 | Agri-Culture 2 | Pasture 1 | Pasture 2 | Native Vegetation |
|---|---|---|---|---|---|---|
| Clay | grams (g)/kilogram (kg) | 518 | 512 | 31 | 496 | 140 |
| Sand | g/kg | 300 | 355 | 926 | 946 | 748 |
| Total organic carbon | g/kg | 33.80 | 29.40 | 0.56 | 0.40 | 14.40 |
| Total porosity | % | 56.60 | 61.71 | 40.23 | 43.54 | 51.51 |
| Aggregate stability index (ASI) | % | 93.45 | 86.93 | 78.59 | 26.59 | 96.06 |
| Mineralogy | n/a | Gibbsite Quartz Kaolinite | Gibbsite Quartz Kaolinite | Gibbsite Quartz | Gibbsite Quartz | Gibbsite Quartz Goethite |
| Slope | % | 5.33 | 1.60 | 13.6 | 3.70 | 5.55 |

### 3.2. WEPP Input Data Parameters

The values for hydraulic conductivity (Ke), interrill soil erodibility (Ki), rill soil erodibility (Kr), and runoff critical shear stress (τc) differed between the watersheds. There were differences between the parameters measured at the sampling points within the watersheds, especially when comparing agriculture and pasture micro-watersheds. Compared to the samples from the agriculture micro-watershed, pasture and native vegetation micro-watersheds had higher Ke, lower Ki, greater Kr, and lower τc (Table 2).

**Table 2.** Hydraulic conductivity ($K_e$), interrill soil erodibility ($K_i$), rill soil erodibility ($K_r$), and runoff critical shear stress ($τ_c$) for the different study areas in the micro-watersheds, Campo Verde, Mato Grosso state, Brazil.

| Micro-Watershed Land Cover | Hydraulic Conductivity ($K_e$) | Interrill Soil Erodibility ($K_i$) | Rill Soil Erodibility ($K_r$) | Runoff Critical Shear Stress ($τ_c$) |
|---|---|---|---|---|
| | Millimeters/Hour | Kilograms/(m × m³) | Seconds/Meter | Newtons/Square Meter |
| Agriculture 1 | $30.63 \pm 6.76$ | $2.44 \times 10^{-5} \pm 0.83 \times 10^4$ | 0.2781 | 0.001483 |
| Agriculture 2 | $24.49 \pm 3.12$ | $13.2 \times 10^{-5} \pm 18.2 \times 10^4$ | 0.1927 | 0.042942 |
| Pasture 1 | $81.52 \pm 15.80$ | $1.56 \times 10^{-5} \pm 2.04 \times 10^4$ | 5.2539 | 0.000949 |
| Pasture 2 | $109.94 \pm 19.34$ | $2.47 \times 10^{-5} \pm 4.14 \times 10^4$ | 11.2723 | 0.000428 |
| Native vegetation | $48.31 \pm 13.41$ | $0.86 \times 10^{-5} \pm 0.21 \times 10^4$ | 1.4383 | 0.002991 |

The $K_e$ and $K_i$ data were reprinted/adapted with permission from Bocuti 2016 [22].

### 3.3. GeoWEPP Model Results

Table 3 shows the sediment production of the watersheds estimated by the GeoWEPP model with the insertion of parameters. Spatial maps of soil depositions and losses are presented for agricultural areas (Figure 5), pasture (Figure 6), and for native vegetation (Figure 7). When simulated without inserting the Ki, Kr, Ke, and τc parameters, the model was unable to accurately estimate the observed soil loss values in the micro-watersheds. However, with the insertion of parameters, soil loss values for micro-basins with pasture and native vegetation were close to the observed values (Table 3). For estimating surface runoff, the model is sensitive only to the hydraulic conductivity parameter (Ke), while the sediment production and soil loss are sensitive to all parameters (Kr, Ke, τc, and Ki) [31].

**Table 3.** Sediment production simulated by GeoWEPP for micro-watersheds in Campo Verde, Mato Grosso state, Brazil.

| Micro-Watershed Land Cover | Observed | | | | Simulated by GeoWEPP Model | |
|---|---|---|---|---|---|---|
| | Total Water Depth Precip-itated (Wdp) | Surface Runoff (SR) | SR Coefficient | Soil Loss | Soil Loss w/NO Calibration Parameters | Soil Loss w/Calibration Parameters |
| | millimeters/year | | % | metric tons/hectares/year | metric tons/hectares/year | metric tons/hectares/year |
| Agriculture 1 Agriculture 2 | 1640.8 | 141.3 | 8.61 | 0.11 | 2.71 2.42 | 5.78 6.16 |
| Pasture 1 Pasture 2 | 1616.8 | 477.8 | 29.55 | 0.06 | 0.28 0.44 | 0.03 0.03 |
| Native vegetation | 1579.3 | 361.8 | 22.91 | 0.10 | 0.21 | 0.10 |

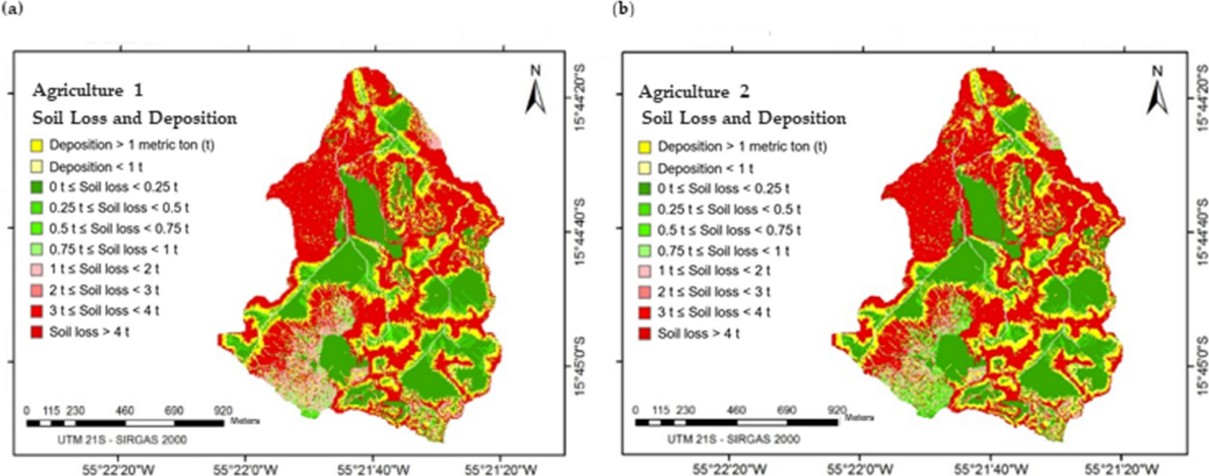

**Figure 5.** Soil gains/losses generated by GeoWEPP for the agricultural cultivation micro-watershed, considering the calibration parameters in Table 2, for sampling points for (**a**) Agriculture 1 and (**b**) Agriculture 2 in the municipality of Campo Verde, Mato Grosso, Brazil.

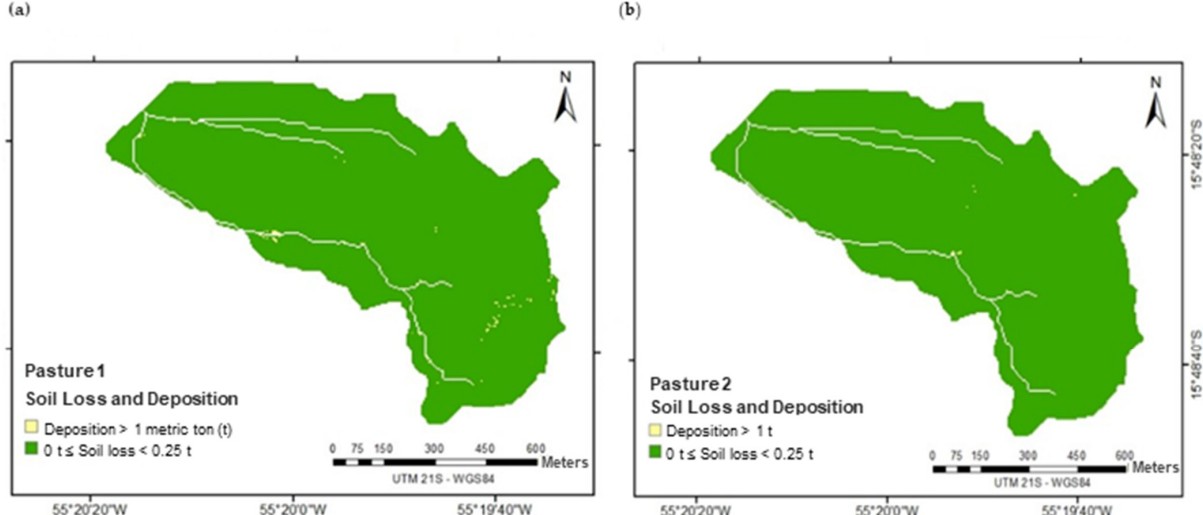

**Figure 6.** Soil gains/losses generated by GeoWEPP fothe micro-watershed for pasture, considering the calibration parameters in Table 2, for the sampling points for (**a**) Pasture 1 and (**b**) Pasture 2 in the municipality of Campo Verde, Mato Grosso, Brazil.

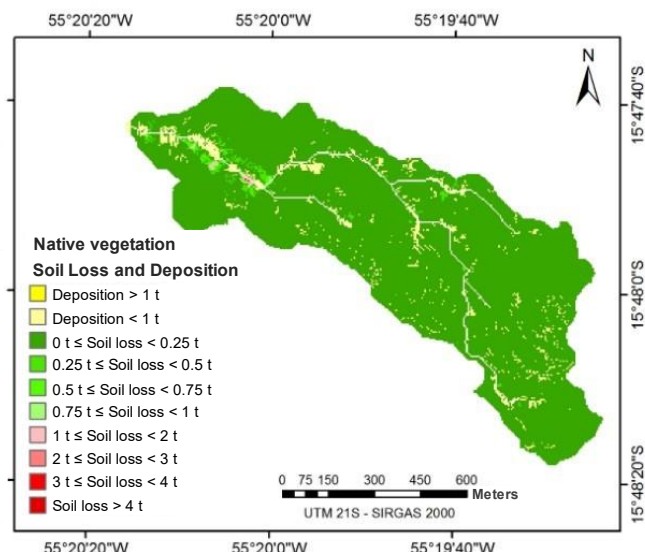

**Figure 7.** Soil gains/losses generated by GeoWEPP for the micro-watershed for native vegetation in the municipality of Campo Verde, Mato Grosso state, Brazil.

According to previous research (Supplementary Materials), the mean runoff rates were lower for areas with higher total sand content due to their higher infiltration capacity and water percolation in these soils [26]. The areas under pasture and native vegetation have a sandy texture, with approximately 3 and 14% clay content, respectively. The area under agricultural production has a clayey texture and, therefore, a higher microporosity and lower macroporosity, leading to less permeability/movement of water in the soil. The model accurately estimated soil losses in the micro-watershed with native vegetation, with values similar to those observed for pasture. The most significant water and soil losses occurred in the micro-watershed with agricultural crops, indicating higher susceptibility to soil loss even after inserting values for interrill erosion ($K_i$), rill erosion ($K_r$), saturated hydraulic conductivity ($K_e$), and soil critical shear stress ($K_e$). However, the surface runoff values decreased significantly. It can be seen that the insertion of Ki, Kr, Ke, and τc data was essential for the main changes that occurred in the modeling of soil loss within micro-basins with pasture and native vegetation, since these parameters are accurately modeled.

The accuracy of the GeoWEPP model in predicting the conditions of sandy soil in the micro-basin with pasture may assist research and extension technicians in predicting erosion and recommending conservation practices. Livestock activity is the main driver of land use change of native vegetation to pasture in the world. In Brazil alone, 60 million hectares have been transformed in this way over the last 33 years [32]. Currently, about 58.8% (equivalent to 97.7 million hectares) of pasture show some degree of degradation [33]. Considering that the GeoWEPP model performed well in estimating soil loss for the area under pasture, it can contribute to making Brazilian meat production more sustainable. Brazil has extensive areas of pasture devoted to extensive grazing of beef cattle, mainly in the Amazon biome but also the Cerrado (i.e., savannah) biome.

The GeoWEPP model overestimated soil losses for agricultural crops with parameterization for the agriculture micro-watershed. The model was developed, and its parameters were adjusted, for temperate climate conditions, which are very different from tropical conditions. Among the three micro-watersheds, crop areas were dominated by a clayey Latossolo Vermelho-Amarelo (Oxisol). In a study of the same micro-watershed, a previous study determined that this soil is basically composed of gibbsite, quartz, and goethite [27], characterized by a granular macrostructure with small loose granules, lower soil density, higher proportion of large pores, and higher permeability [34]. This implies these soils are less erodible compared to Oxisols with kaolinitic mineralogy, for example [35]. A prior study estimating the interrill erodibility of Oxisols in Rio Grande do Sul found that the WEPP model overestimated experimental values by 2.4 to 3.6 times [36]. These researchers

attributed this overestimation to the equations used in the WEPP model. American soils are generally sandier, and WEPP was developed based on soils subject to a lower weathering rate compared to Brazilian Oxisols.

Although the GeoWEPP model overestimated the values of soil loss in the micro-watershed with agriculture, it can be seen that the practices used in this area have contributed to the maintenance and conservation of soil and water. The observed values of soil loss in this system are 0.11 metric tons/hectare/year, which is considered below the threshold considered suitable for tropical soils. The existence of terracing in the agricultural areas we modeled may also have contributed to the WEPP model not being able to successfully estimate soil loss. This occurs because terraces smaller than 5 m may not have been captured by the digital elevation model (DEM) during mapping.

Our erosions measurements in the field for crops and pasture were consistent with another recent study measuring erosion with simulated rainfall in both the Renato sub-basin and Caiabi sub-basin in the middle and upper regions of Teles Pires River basin in Mato Grosso state, Brazil, respectively. These sub-basins are in the Amazon-Cerrado transition zone, which is north of where we conducted our experiment in the southern Mato Grosso state Cerrado biome. The types of soil in this Teles Pires study's experimental area were all Latisols (versus Oxisols). For crops with crop residue, these researchers measured erosion ranging from 0.035 to 0.102 metric tons (t)/hectare (ha) [37]. This is slightly less than what we measured at 0.11 t/ha (Table 3). The erosion we measured for pasture at 0.06 t/ha (Table 3) was within the range measured for erosion in pasture (0.0101 to 0.087 t/ha) by Alves et al. 2023 [37].

Some studies have observed that anthropogenic factors such as land use and management have more influence on soil loss than precipitation and topography [38]. This is consistent with the results from our study, as evidenced by the hydrological dynamics and surface runoff in the micro-watersheds. A study in the same micro-watersheds we studied that were in pasture and native vegetation found changes in the balance of carbon and nutrients, which were attributed to anthropic factors [39].

The WEPP model indicated lower soil losses in areas with pasture and native vegetation. This result was already expected since the experimental results obtained in the field showed high hydraulic conductivity. The experimental units with pasture had 81.52 and 109.94 mm (mm) of effective soil hydraulic conductivity, while the area with native vegetation had 48.31 mm [25]. Thus, the calibrated model gave these areas the lowest potential to generate soil losses, especially when considering land use and vegetative cover factors. Hydraulic conductivity is one of the parameters in which the GeoWEPP model is more sensitive [40], which can considerably affect model results, as lower water infiltration corresponds to a higher flow rate and greater sediment transport capacity [41].

The characteristics observed in the micro-watersheds we modeled indicate the extent to which soil erosion is triggered since these characteristics describe the processes related to morphometric characteristics and land use and management. These processes are measured by the amount of material removed from the soil per unit area and per unit time. Such surface runoff happens when the water volume is higher than the infiltration limit, which leads to runoff [42]. The processes of transport, deposition, and sedimentation of soil particles occur more intensely in naturally exposed land or under intensive management practices such as those for agriculture, which can lead to landscape degradation.

Table 4 shows the values of the root mean square error (RMSE), the concordance index (d), and the Nash–Sutcliffe coefficient (NS). The NS values were negative for all areas. Wilmott's concordance index (d) expresses the accuracy of the estimates in relation to the observed values. It appears that the model presented a higher level of accuracy for the pasture micro-watershed, presenting the value closest to unity (d = 0.56). Thus, it appears that the WEPP model presented more accurate measurements when applied to watersheds with sandy characteristics, which is corroborated by the work of Amorim et al. (2010), in which the worst performances of the WEPP model occurred in plots conducted in Latossolos (Oxisols) [17].

**Table 4.** Statistical parameters for the evaluation of soil loss estimates by the GeoWEPP model for the Brazilian Cerrado with insertion of calibration parameters.

| Micro-Watershed | Statistical Parameters | | |
|---|---|---|---|
| | Root Mean Squared Error (RMSE) | Willmott Concordance Index (d) | Nash-Sutcliffe (NS) |
| Agriculture | 0.12 | 0.27 | −3.38 |
| Pasture | 0.48 | 0.59 | −0.56 |
| Native vegetation | 0.68 | 0.44 | −0.59 |

Figure 8 shows the relationship between the soil loss estimates using the parameterized GeoWEPP model compared to the same model without the parameterization. Most of the points for the micro-watersheds with pasture and native vegetation are located below the zero deviation line, indicating the tendency for underestimation of soil losses by the model for the land use and management conditions in GeoWEPP simulations. The main changes with the parameterization occurred for agricultural production in the micro-watershed. In general, the USLE, RUSLE, and WEPP erosion models tend to overestimate small events and underestimate large events [43]. This feature is even more evident in the WEPP model [44], and the model may not be able to estimate or correct soil loss values. A possible explanation for unsatisfactory results for model runs simulated by GeoWEPP may be related to the way GeoWEPP analyzes events. GeoWEPP analyzes runoff and sediment generation daily. The response time for a rainfall event may last only a few hours. Thus, the error for events could be minimized if GeoWEPP analyzed the rainfall events with a minimum time step of one hour. The approximations performed for land cover or even the calibration parameters to represent natural phenomena may have caused errors or may not be representing all the phenomena that impact soil erosion.

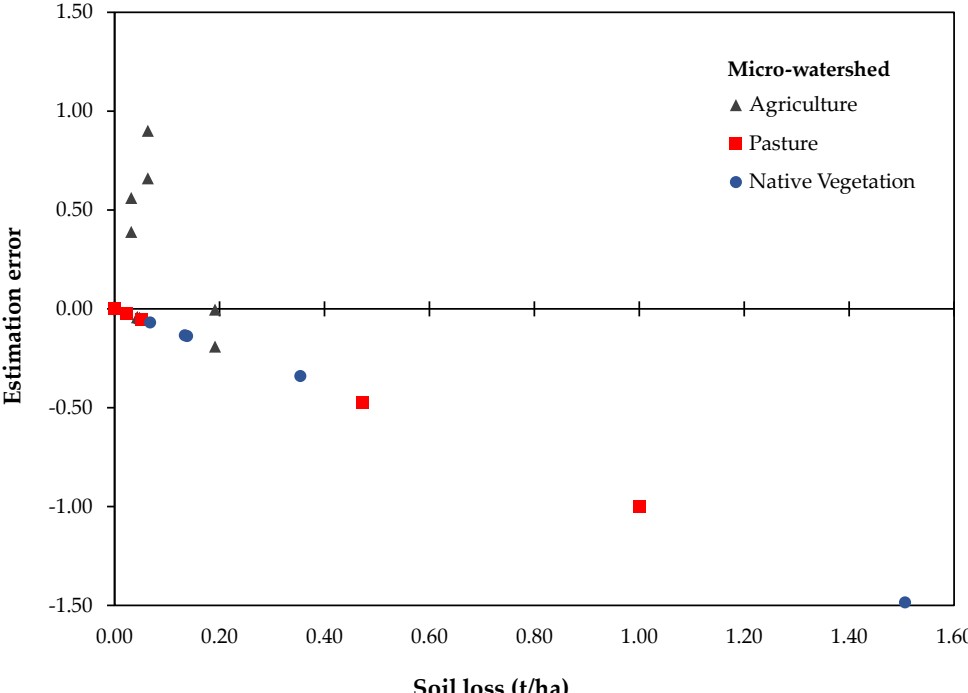

**Figure 8.** Estimation error of soil loss by the GeoWEPP model versus observed soil losses measured in metric tons (t) per hectare (ha) under land use and management conditions in the micro-watersheds for agriculture, pasture, and native vegetation in Campo Verde, Mato Grosso state, Brazil.

The determination of the minimum length of drainage networks and the minimum critical area by GeoWEPP may also have had a considerable influence on our results. This is because the greater the drainage density, the faster that water and sediments are drained by the basin, which can generate larger peaks. Limited research in Brazil has validated GeoWEPP using calibration parameters from multi-year field data. However, several of these researchers point out that the model has a high level of uncertainty due to the complexity in evaluating soil erosion [45].

It can be Inferred that the inclusion of the calibration parameters for hydraulic conductivity ($K_e$), interrill erodibility ($K_i$), rill erodibility ($K_r$), and critical shear stress ($\tau_c$) alter the flow dynamics and sediment production in the outlet generated by the GeoWEPP model. Therefore, the estimation of these parameters by WEPP can lead to unexpected results. This is because the model has numerous parameters for land use characteristics, but all for American soils under temperate climate conditions.

Prior research found that the use of parameters obtained experimentally improved on average, 306% of the estimates for soil loss, when compared to the use of Ke, Ki, Kr, and τc, generated internally by WEPP. The same occurred for water losses, providing an improvement of 135% [46]. When analyzing the sensitivity of several input parameters of the WEPP model for the Brazilian edaphoclimatic conditions, a previous study verified that the Ke, Ki, Kr, and τc parameters were the ones that presented the greatest improvements within the model [47].

### 3.4. Future Directions for Sustainable Agricultural Development

The two types of soils represented in our analyses make up over half of the land area of Brazil (Oxisols 38.7 + Entisols 14.5 = 53.2%) [48]. Although the GeoWEPP model validation for crops requires future improvement, the pasture and natural vegetation we validated the model for make up a large percentage of the Cerrado savannah biome at 54.8% for such natural areas and 30.7% for pasture with crops taking up only 13.1% of the land area of the Cerrado biome [49]. Future studies using GeoWEPP can validate the model across a greater variation in types and density of natural vegetation cover since it has been shown in more arid environments such as Brazil's northeast region that denser vegetation cover is associated with lower erosion metrics and greater soil carbon [50]. Natural vegetation buffers in the Cerrado confer ecosystem service benefits such as biodiversity and improved soil and water quality [51].

GeoWEPP can be used to identify priority pasture areas in Brazil for improved soil conservation. Soil erosion is a significant challenge in Brazil, especially in agricultural areas with sandy soils, which can lead to noticeable erosion as rills and gullies even in areas with pasture [52], as well as increased suspended sediments loads in rivers such as the Teles Pires [53]. Brazil's pastures help support ~253 million head of cattle [49]. Using GeoWEPP to model erosion in Brazil's pastures is important due to the predominance of the extensive grazing system in Brazil and the susceptibility of this type of grazing to pasture degradation. Of Brazil's ~850 million hectares or ha (~8.5 million square kilometers or $km^2$) of total land area, about 21% of this is in pasture or ~179 million ha (~1.79 million $km^2$). About 35.7% of Brazil's pasture is degraded or 63.7 million ha (637,000 $km^2$) [54]. Therefore, using GeoWEPP to help rehabilitate degraded pastures is paramount to improving large ruminant livestock productivity and profitability, which can produce more cattle on less land. Such land sparing can allow for the restoration of natural habitats.

## 4. Conclusions

The GeoWEPP model performs well in estimating soil loss for areas under pasture and native vegetation; however, further study is warranted to increase the model's accuracy in estimating soil loss for areas where there is annual cultivation of commodity crops such as soybeans and maize. The GeoWEPP model performed better when the calibration parameters for both interrill and rill soil erodibility, soil critical shear stress, and saturated hydraulic conductivity were used in the model. The model slightly underestimated soil loss

in pasture while accurately modeling this for native vegetation. GeoWEPP overestimated erosion in the micro-watershed we evaluated for agricultural production. In general, the GeoWEPP model can be used in a predictive way if there is an available digital elevation model for the terrain being modeled. This model can facilitate planning by agricultural technicians and by state, university, and independent researchers. As a whole, it is an important predictive tool for soil and environmental management in this region. This is due to GeoWEPP's ability to generate numerous scenarios of land use and management, which can help predict possible impacts from erosion. Future studies should focus on better calibrating the model with commodity cropping systems and expanding the use of this model to other tropical areas in Brazil, as well as around the world.

**Supplementary Materials:** Related supplementary material can be found in previous research at: (1) https://doi.org/10.5433/1679-0359.2020v41n5Supl1p1909 [19], (2) https://doi.org/10.1371/journal.pone.0179414 [23], (3) https://doi.org/10.19084/RCA18130 [25], (4) http://dx.doi.org/10.1590/1807-1929/agriambi.v24n6p357-363 [41], (5) https://doi.org/10.1016/j.gecco.2019.e00819 [51].

**Author Contributions:** Conceptualization, W.d.A.M. and R.S.S.A.; methodology, W.d.A.M., W.M.d.S. and E.D.B.; software, W.d.A.M.; validation, W.d.A.M., R.S.S.A. and E.D.B.; formal analysis, W.d.A.M., R.S.S.A., A.K.H. and W.M.d.S.; investigation, E.D.B.; resources, R.S.S.A., E.D.B., M.O.H., A.K.H. and W.d.A.M.; data curation, M.O.H. and D.C.d.A.; writing—original draft preparation, W.d.A.M., W.M.d.S., A.K.H. and M.O'H.H; writing—review and editing, W.d.A.M., A.K.H., D.C.d.A., L.A.D.L.D.R. and W.M.d.S.; visualization, L.A.D.L.D.R., A.K.H.; D.C.d.A. and W.M.d.S.; supervision, M.O.H. and R.S.S.A.; project administration, R.S.S.A. and E.D.B.; funding acquisition, R.S.S.A. All authors have read and agreed to the published version of the manuscript.

**Funding:** This research was developed in partnership with the CarBioCial project (donation number: 01 LL0902A). The authors are also grateful for financial support from the Mato Grosso State Research Support Foundation (www.fapemat.mt.gov.br, accessed on 5 December 2019), grant number: 335908/2012 and the Brazilian National Council for Scientific and Technological Development (www.cnpq.br, accessed on 11 December 2019), grant number: 481990/2013-5.

**Institutional Review Board Statement:** Not applicable.

**Informed Consent Statement:** Not applicable.

**Data Availability Statement:** Study data can be obtained by request to the corresponding author or the first author via e-mail.

**Acknowledgments:** The authors would like to thank Fazenda Santa Luzia and Fazenda Rancho do Sol for providing the study areas, and CAPES for granting the doctoral scholarship to the first author. The authors also acknowledge the collaboration of field hosts (Fazendas Santa Luzia and Rancho do Sol) and the field assistance of Túlio G. Santos and Alan R.R. Martin. We thank four anonymous reviewers whose comments and edits improved the quality of our work.

**Conflicts of Interest:** The authors declare no conflict of interest. Supporting entities had no role in the design of the study; in the collection, analyses, or interpretation of data; in the writing of the manuscript, or in the decision to publish the results.

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
