# Peer review of "Using the GeoWEPP Model to Predict Water Erosion in Micro-Watersheds in the Brazilian Cerrado"

_sustainability, doi:10.3390/su15064711_

Round 1
Reviewer 1 Report
The following small revisions are necessary.
1) Figure 1, Left ; The Native Vegetation Watershed is not visible.
2) Figure 4; What is “WEPP”? What is the deference between WEPP and GeoWEPP?
3) What is the difference between AGRICULTURE 1 and 2?
4) In Figure 6, the error increases when the soil loss increases. This means that the error does not convert to the small value. Please have a comment about the error in case that the soil loss becomes larger than 1.5 (t/ha).
Author Response
Comments and Suggestions for Authors
The following small revisions are necessary.
1) Figure 1, Left ; The Native Vegetation Watershed is not visible.
It is in green color, above the grazing basin.
2) Figure 4; What is “WEPP”? What is the deference between WEPP and GeoWEPP?
GeoWEPP is an ArcView 3.x or ArcGIS 9.x extension that allows the use of a digital elevation model (DEM) and topographic maps to generate watershed configurations, together with the parameterization files generated in WEPP (climate, soil , management and topography).
Text inserted in the manuscript on lines 253-258 to clarify this.
3) What is the difference between AGRICULTURE 1 and 2?
These are soil sampling points within micro-watersheds, for determining parameters and estimating soil loss by the model. A supplement was inserted in the text, in section 2.2.:
“To carry out the experimental analyzes (soil and water loss), two representative sampling points were selected within the watershed under pasture and agriculture use, and one point in the area under native vegetation, described in the paper [17].”
4) In Figure 6, the error increases when the soil loss increases. This means that the error does not convert to the small value. Please have a comment about the error in case that the soil loss becomes larger than 1.5 (t/ha).
Correction applied to the text on lines 488-490.
Submission Date
02 February 2023
Date of this review
13 Feb 2023 22:35:48
Reviewer 2 Report
The paper titled Using the GeoWEPP Model to Predict Water Erosion in Micro-Watersheds in the Brazilian Cerrado represents a major contribution to the field of environmental impact, soil conservation, and water erosion prediction using the GeoWEPP Model. The authors successfully evaluated the performance of the GeoWEPP model in estimating soil erosion in three micro-watersheds in the savannah of southeastern Mato Grosso state.
This topic is very important, original, and relevant in this field of research. The methodology used is adequate for this type of research. The paper highlighted the importance of using this model for several reasons (for example, due to the predominance of the extensive grazing system in Brazil and the susceptibility of this type of grazing to pasture degradation).
Quotations are relevant, and the references are appropriate. The research design is appropriate, and the methods are adequately described. The results are presented adequately and comprehensibly. The authors gave the future directions for sustainable agricultural development which is very important for future studies. All figures and tables are well-presented, but the authors could label the figures more clearly. For example, Figure 2 contains 4 photos and Figure 3 - 2 photos. Figures 2 should be named Figures 2-5, or Figures 2a-d. If the numbering of Figures is corrected, the same should be corrected in the text.
The conclusion is supported by the results, and they are consistent with the evidence and arguments presented in the manuscript.
Some technical issue: At the conclusion between the first and second sentences there is an unnecessary sign -. Please correct.
The article is acceptable for publication in Sustainability after minor revision.
Author Response
Comments and Suggestions for Authors
The paper titled Using the GeoWEPP Model to Predict Water Erosion in Micro-Watersheds in the Brazilian Cerrado represents a major contribution to the field of environmental impact, soil conservation, and water erosion prediction using the GeoWEPP Model. The authors successfully evaluated the performance of the GeoWEPP model in estimating soil erosion in three micro-watersheds in the savannah of southeastern Mato Grosso state.
This topic is very important, original, and relevant in this field of research. The methodology used is adequate for this type of research. The paper highlighted the importance of using this model for several reasons (for example, due to the predominance of the extensive grazing system in Brazil and the susceptibility of this type of grazing to pasture degradation).
Quotations are relevant, and the references are appropriate. The research design is appropriate, and the methods are adequately described. The results are presented adequately and comprehensibly. The authors gave the future directions for sustainable agricultural development which is very important for future studies.
All figures and tables are well-presented, but the authors could label the figures more clearly. For example, Figure 2 contains 4 photos and Figure 3 - 2 photos. Figures 2 should be named Figures 2-5, or Figures 2a-d. If the numbering of Figures is corrected, the same should be corrected in the text.
Corrections applied to these figures as requested.
The conclusion is supported by the results, and they are consistent with the evidence and arguments presented in the manuscript.
Some technical issue: At the conclusion between the first and second sentences there is an unnecessary sign -. Please correct.
Correction applied
The article is acceptable for publication in Sustainability after minor revision.
Submission Date
02 February 2023
Date of this review
09 Feb 2023 09:22:28
Reviewer 3 Report
This work evaluated the performance of the GeoWEPP model in estimating sediment production from three micro-watersheds of the Cerrado in Mato Grosso state, Brazil. The topic of this manuscript is interesting to the readers of the journal. Per the comparison of the modelling results with filed observations, the authors concluded GeoWEPP best modeled soil erosion for native vegetation and pasture, but over-estimated that for crops. Surface runoff was best modeled for crops versus native vegetation and pasture. While these observations are helpful in “sustainability”, a stronger work should also demonstrate and/or suggest what the key parameters are and how to adjust such key parameters (i.e., through relevant filed management/operations) to meet the sustainability challenges in the modelling regions. Authors should elaborate this point more in abstract and main text.
In addition, extra English polishing is encouraged. For example, in the beginning of the abstract, the singular “GeoWEPP model” is inconsistent with the plural “these models” of the second sentence (Lines 24-25).
L66. Seems citation(s) needed after “soil loss”.
L489, what is “out analyses”? Or is it “our anlysese”?
Author Response
Comments and Suggestions for Authors
This work evaluated the performance of the GeoWEPP model in estimating sediment production from three micro-watersheds of the Cerrado in Mato Grosso state, Brazil. The topic of this manuscript is interesting to the readers of the journal. Per the comparison of the modelling results with filed observations, the authors concluded GeoWEPP best modeled soil erosion for native vegetation and pasture, but over-estimated that for crops. Surface runoff was best modeled for crops versus native vegetation and pasture. While these observations are helpful in “sustainability”, a stronger work should also demonstrate and/or suggest what the key parameters are and how to adjust such key parameters (i.e., through relevant filed management/operations) to meet the sustainability challenges in the modelling regions. Authors should elaborate this point more in abstract and main text.
Correction applied to the abstract
In addition, extra English polishing is encouraged. For example, in the beginning of the abstract, the singular “GeoWEPP model” is inconsistent with the plural “these models” of the second sentence (Lines 24-25).
Correction applied to the text.
L66. Seems citation(s) needed after “soil loss”.
We have cited this on line 67.
L489, what is “out analyses”? Or is it “our anlysese”?
This is a typographical error. We corrected for “our” analyses
Submission Date
02 February 2023
Date of this review
10 Feb 2023 16:59:34
Reviewer 4 Report
In my opinion this paper is an interesting study and authors have evaluated the performance of the GeoWEPP model in estimating sediment production, which applied to three micro-watersheds of the Cerrado in Mato Grosso state, Brazil.
Dear authors, thank you for submitting the manuscript to the journal of Sustainability. Its topic is very interesting. However, the current version of the paper suffers from a number of weaknesses related to the empirical strategy used. I have the following comments/questions for the authors:
Abstract
· Don’t use abbreviation on the first time. Define full form for the first time than after use abbreviation only (Please check in the entire manuscript).
· Add important results in the abstract section.
· The authors ought to re-write the abstract so that it briefly presents the problem at hand, objectives of the study, methods used to achieve the objectives in logical order. Also, abstract section should be completed with the results of the study.
Introduction
· Try to highlight the regional or national significance of this study.
· In introduction chapter please focus on problem generally, on the basis of examples in the whole World, not your study area.
· Please, transfer Figure 1 under study area (Materials and Methods section).
· Add some recent article to make your introduction more attractive and strong. I propose to add this survey method in the overview section of the introduction section, based on the latest literature. Please replace old citations (if it is possible) or add citations of newest literature.
https://doi.org/10.3390/w14060890
https://doi.org/10.3390/w13213094
Materials and Methods
Study area
· Describe all the features of the study area in brief including climate, topography, geology, and hydrogeology?
Soil Charactierization & Soil Sampling
· Please give detailed information on soil samplers (e.g., accuracy, manufacturer).
· Sampling locations were selected carefully within the three selected micro-watersheds. What criteria where analyzed to select this locations?
· Please provide detailed detection methods and quality control results?
· Please support your methods by providing appropriate references or give the guidelines used and equations.
Results and Discussion
· You should think how transformational the research is likely to be should be made so that the outcome of the work will have an impact on the community/society facing given sustainability related challenges?
· Write the practical applications of your work in a separate section, before the conclusions and provide your good perspectives.
· What are the economic benefits of this work?
· What long-term impacts will it have on environmental protection and the wider public or the field following the completion of the research?
Conclusion
· Concise the text in conclusion and add future work in order to recommend your work. Shorten the length of each and every paragraph by adding only relevant and major findings in your study.
Please respond to all of those comments in the revised manuscript by pointing out precisely and concisely on which page and in which line you have incorporated your response one by one.
Author Response
Comments and Suggestions for Authors
In my opinion this paper is an interesting study and authors have evaluated the performance of the GeoWEPP model in estimating sediment production, which applied to three micro-watersheds of the Cerrado in Mato Grosso state, Brazil. Dear authors, thank you for submitting the manuscript to the journal of Sustainability. Its topic is very interesting. However, the current version of the paper suffers from a number of weaknesses related to the empirical strategy used. I have the following comments/questions for the authors:
Abstract
Don’t use abbreviation on the first time. Define full form for the first time than after use abbreviation only (Please check in the entire manuscript).
Correction applied to the text.
Add important results in the abstract section.
Correction applied to the abstract, on lines 25-27.
The authors ought to re-write the abstract so that it briefly presents the problem at hand, objectives of the study, methods used to achieve the objectives in logical order. Also, abstract section should be completed with the results of the study.
Improvements made in order to meet the reviewer's suggestions on lines 25-27.
Introduction
Try to highlight the regional or national significance of this study.
Correction applied to the text, on lines 94-97.
In introduction chapter please focus on problem generally, on the basis of examples in the whole World, not your study area.
Correction applied to the text, on lines 43-44.
Please, transfer Figure 1 under study area (Materials and Methods section).
We have moved Figure 1 from the Introduction section to the Methods section.
Add some recent article to make your introduction more attractive and strong. I propose to add this survey method in the overview section of the introduction section, based on the latest literature. Please replace old citations (if it is possible) or add citations of newest literature.
https://doi.org/10.3390/w14060890
https://doi.org/10.3390/w13213094
Some more recent papers on the subject have been added to the manuscript. Discussions and references inserted on line 458 and lines 487-490, in order to improve the importance of the theme.
Materials and Methods
Study area
Describe all the features of the study area in brief including climate, topography, geology, and hydrogeology?
Correction applied to the text, on lines 117-127.
Soil Charactierization & Soil Sampling
Please give detailed information on soil samplers (e.g., accuracy, manufacturer).
Correction applied to the text, on lines 143-149.
Sampling locations were selected carefully within the three selected micro-watersheds. What criteria where analyzed to select this locations?
Correction applied to the text: “The criteria used were: local relief, soil color and texture, history of use and occupation.”
Please provide detailed detection methods and quality control results?
The methods are based and foundations on previously published work, and this work is a sequence of other works developed in the same areas. The applied equations are based on the publication of other published papers on the WEPP model.
Please support your methods by providing appropriate references or give the guidelines used and equations.
The methods are based and foundations on previously published work, and this work is a sequence of other works developed in the same areas. The applied equations are based on the publication of other published papers on the WEPP model.
Results and Discussion
You should think how transformational the research is likely to be should be made so that the outcome of the work will have an impact on the community/society facing given sustainability related challenges?
We summarized the importance of the results of the work in the topic “3.4. Future Directions for Sustainable Agricultural Development.”
Write the practical applications of your work in a separate section, before the conclusions and provide your good perspectives.
We followed your suggestions in the topic “3.4. Future Directions for Sustainable Agricultural Development.”
What are the economic benefits of this work?
We have clarified this on lines 545-546.
What long-term impacts will it have on environmental protection and the wider public or the field following the completion of the research?
We have clarified this on lines 546-547.
Conclusion
Concise the text in conclusion and add future work in order to recommend your work. Shorten the length of each and every paragraph by adding only relevant and major findings in your study.
Adjustment made in order to be more objective in each paragraph and we have added future work on lines 549-564.
Please respond to all of those comments in the revised manuscript by pointing out precisely and concisely on which page and in which line you have incorporated your response one by one.
Submission Date
02 February 2023
Date of this review
09 Feb 2023 00:39:59
Round 2
Reviewer 1 Report
The test is well revised. I hope that the error indicated in Figure 6 becomes smaller in the future research.
Reviewer 2 Report
-
Reviewer 3 Report
The authors did not address my major concern on “a stronger work should also demonstrate and/or suggest what the key parameters are and how to adjust such key parameters”, just claiming English is polished while they responded to the main comment.
I am OK if the editor decides to accept the current manuscript for publication.
Reviewer 4 Report
In my opinion this paper is an interesting study and authors have evaluated the performance of the GeoWEPP model in estimating sediment production, which applied to three micro-watersheds of the Cerrado in Mato Grosso state, Brazil.
The article is written correctly, includes a discussion of the research findings, and a good review of the literature. The results are presented in a clearly structured manner. The manuscript has been significantly improved and can now be accepted in current form.